# A Rigorous Observation Model for the Risley Prism-Based Livox Mid-40 Lidar Sensor

**DOI:** 10.3390/s21144722

**Published:** 2021-07-10

**Authors:** Ryan G. Brazeal, Benjamin E. Wilkinson, Hartwig H. Hochmair

**Affiliations:** 1Geomatics Program, School of Forest, Fisheries, and Geomatics Sciences, University of Florida, Gainesville, FL 32611, USA; ryan.brazeal@ufl.edu; 2Geospatial Modeling and Applications Laboratory, School of Forest, Fisheries, and Geomatics Sciences, University of Florida, Gainesville, FL 32611, USA; 3Geomatics Program, Fort Lauderdale Research & Education Center, School of Forest, Fisheries, and Geomatics Sciences, University of Florida, Fort Lauderdale, FL 33314, USA; hhhochmair@ufl.edu

**Keywords:** lidar, laser scanning, Risley prism, calibration, UAS, drones

## Abstract

Modern lidar sensors are continuing to decrease in size, weight, and cost, but the demand for fast, abundant, and high-accuracy lidar observations is only increasing. The Livox Mid-40 lidar sensor was designed for use within sense-and-avoid navigation systems for autonomous vehicles, but has also found adoption within aerial mapping systems. In order to characterize the overall quality of the point clouds from the Mid-40 sensor and enable sensor calibration, a rigorous model of the sensor’s raw observations is needed. This paper presents the development of an angular observation model for the Mid-40 sensor, and its application within an extended Kalman filter that uses the sensor’s data to estimate the model’s operating parameters, systematic errors, and the instantaneous prism rotation angles for the Risley prism optical steering mechanism. The analysis suggests that the Mid-40’s angular observations are more accurate than the specifications provided by the manufacturer. Additionally, it is shown that the prism rotation angles can be used within a planar constrained least-squares adjustment to theoretically improve the accuracy of the angular observations of the Mid-40 sensor.

## 1. Introduction

Small form factor lidar sensors have become commonplace within numerous research areas and commercial applications, including autonomous vehicle navigation [1,2,3], mapping from unoccupied aerial systems (UAS), as known as remotely piloted aircraft systems (RPAS) [4,5,6], terrestrial-based personal mobile mapping systems [7,8], and recently for improving imaging and augmented reality capabilities on mobile devices [9,10]. Before integrating a lidar sensor into a remote sensing platform it is fundamental to understand the capabilities of the sensor, and this can only be achieved by a systematic evaluation of the accuracy, repeatability, and stability of the sensor [11]. Most commercially available, small form factor lidar sensors have already been systematically evaluated by researchers and the results published within literature (e.g., [11,12,13,14]). However, the Livox Mid-40 lidar sensor (Figure 1a)—which became commercially available in 2019—has only been partially evaluated to date, as its angular observations have not been rigorously evaluated. The principal topics of this study are the angular observations of the Mid-40 sensor and its suitability for use within UAS-based lidar mapping applications.

Unlike a multi-line lidar sensor (e.g., Velodyne VLP-16) which uses a mechanical rotating array of multiple laser transceivers, the Mid-40 uses only a single laser transceiver that is in a stationary position and orientation relative to the sensor’s own coordinate system, similar to most terrestrial laser scanners (TLS). However, unlike TLS which commonly use mirrors to reflect the laser beam, the beam of the Mid-40 is steered using a Risley prism optical mechanism consisting of two wedge-shaped glass prisms aligned sequentially in the direction of the optical scan axis (Figure 1b). Each prism can rotate about the optical scan axis resulting in the incident beam being refracted by the glass prisms. The emergent beam is deviated in a direction according to the relative orientation of the prisms with respect to each other [15].

Using a Risley prism mechanism allows for a wide range of unique scan patterns to be realized simply by adjusting the rotation rates and directions of each prism, see [15] for complete details. The scan pattern observed by the Mid-40 is commonly described as a rosette pattern and it occurs within a circular field of view (FoV) around the optical scan axis (Figure 2a). The rotation rates for the glass prisms inside the Mid-40 are set so the lidar observations are not repetitively taken along a single projected path within the FoV, rather the observation density within the FoV increases with time (Figure 2b). The observation density within the FoV is nonhomogeneous with a peak at the center, similar to the response of the human retina [16].

In comparison to the laborious manufacturing techniques used for mechanical multi-line lidar sensors, the Mid-40 is mass produced using a straightforward approach which enables the sensor to be sold at a lower cost [16]. In addition to the lower cost, the reported accuracy specifications and physical properties of the sensor make it an attractive option for UAS lidar mapping applications (Table 1). However, in order to fully understand the quality of the point clouds produced by the Mid-40, an assessment of the sensor’s accuracy specifications is needed. The primary research of this study is a rigorous examination of the systematic errors that affect the raw angular observations (i.e., azimuth angle and zenith angle) of the Livox Mid-40 lidar sensor for the purposes of accuracy assessment.

To date, two studies have been published that provide an initial assessment of the accuracies of the Mid-40. In Ortiz Arteaga et al. [18], the range and angular observation accuracies of the sensor were examined by comparing features within the collected point clouds (i.e., signalized targets and planes) to the same features within a more precise reference point cloud collected by a TLS. The study concluded that at longer standoff distances—40 m to 130 m—the Mid-40 achieved range measurements to an accuracy of ±20 mm. The angular accuracy was inferred by a trigonometric approximation of the ratio of the observed length error between known signalized targets over the standoff distance between the Mid-40 and the targets. The study concluded the angular accuracy of the Mid-40 to be 0.1°, but reported that an angular systematic effect may be present based on the appearance of concentric “ripples” of noise artifacts within some of the collected point clouds. The artifacts were believed to be correlated with the deflection angle of the emergent beam with respect to the optical scan axis and appeared within point clouds of planar surfaces that were observed from small standoff distances.

In Glennie and Hartzell [19], the range and angular observation accuracies were again examined by comparing the collected point cloud datasets to a more precise reference point cloud from a TLS. The analysis included 127 planar comparisons collected from standoff distances between 3 m to 35 m, and from incidence angles between approximately 5° and 85°. The study concluded that the Mid-40 exhibited two classifications of range accuracies, ±8 mm for range observations greater than 20 m and ±21 mm for range observations less than 20 m. The angular accuracy was not explicitly analyzed, but a visual examination to identify any systematic effects correlated to the angular observations was conducted, see [19] for complete details. An attempt was made to duplicate the systematic “ripple effect” reported in [18], but no point cloud artifacts were found. The study concluded that no obvious systematic trends in the angular observations were observed.

Refs. [18,19] quantified the range accuracy of the Mid-40 to be in agreement with the manufacturer’s specification of ±20 mm and provided some evidence that the angular accuracy agrees with the manufacturer’s specification of <0.1°. However, both studies stated that the lack of information regarding the internal hardware configuration and operating principles of the Mid-40 restricted the ability to perform further analysis and uncover additional systematic errors.

This study presents the development of an angular observation model for the Livox Mid-40 lidar sensor. The observation model captures the physical prism parameters and systematic errors that affect the incident beam as it travels and refracts through the Risley prism steering mechanism and exits the sensor as the observable emergent beam. The observation model is used within an extended Kalman filter (EKF) to suitably represent the physical dynamics of the sensor and to statistically estimate the model parameters and the rotation angles for the individual prisms that correspond to the sequential azimuth and zenith observations. The filter’s inputs are the azimuth and zenith observations output directly by the Mid-40 sensor.

An initial analysis is performed using actual observations from a Mid-40 sensor to quantify the accuracy of the angular observation model and provide realistic estimates for the systematic errors. A secondary analysis is performed using simulated input observations—with a known amount of noise—to illustrate the filter’s ability to estimate the systematic errors and the prism rotation angles accurately and precisely. A final analysis is performed using simulated observations from an incorrectly calibrated sensor to a planar surface, to illustrate that the angular observations from an incorrectly calibrated sensor can be improved by using the range observations and applying a planar constraint within a least-squares adjustment. Lastly, a discussion on possible systematic errors within the range observation and the Cartesian coordinates representation of the observations is presented.

## 2. Materials and Methods

When discussing a conventional Risley prism steering mechanism there are four possible configurations for the internal arrangement of the two wedge-shaped glass prisms. Each prism has a perpendicular face (P) and an angled face (A) with respect to the optical scan axis. Therefore, the four configurations are PA-PA, PA-AP, AP-AP and AP-PA (Figure 3). Though each configuration follows the same generalized mathematical form for the direction of the emergent beam, selecting the correct configuration for the specific Risley prism mechanism being studied is important. Based on the information presented in [16], the Mid-40 Risley prism configuration is PA-AP.

### 2.1. Ideal Angular Observation Model

The angular observation model for a Risley prism steering mechanism starts with the unit vector of the incident beam and computes four successive refractions—using the 3D vector form of Snell’s law (Equation (10))—at the perpendicular and angled faces of each prism. Under ideal conditions, the incident beam, the planar normal on the perpendicular face of prism A, the axis of rotation for both prisms, and the planar normal on the perpendicular face of prism B are all collinear with the optical scan axis. The deflection angle between the perpendicular planar normal and the angled planar normal is equal to the wedge angle of the prism. At the zero position of the prisms, all four planar normals for the faces of the prisms lie in a common vertical plane (Figure 4). The prisms are geometrically identical and made of the same homogeneous medium with a precise refractive index. The prisms rotate at constant angular velocities and the rotation angles for each prism—with respect to the zero position—at the time of observation are expressed by:(1)$$\Omega_{A}=\omega_{A}\times t$$
(2)$$\Omega_{B}=\omega_{B}\times t$$
where $\Omega_{A}$ and $\Omega_{B}$ represent the rotation angles for prism A and prism B at the time of observation t, respectively, and $\omega_{A}$ and $\omega_{B}$ represent the constant angular velocities for prism A and prism B, respectively.

The reference coordinate system for the ideal observation model is defined as a right-handed system with its origin at the intersection of the axis of rotation of prism B and the perpendicular face of prism B. The optical scan axis defines the X coordinate axis and the common vertical plane contains the orthogonal Z coordinate axis (Figure 4). The unit vectors for the direction of the incident beam ($L_{0}$), the planar normals for all four prism faces ($N_{1}$, $N_{2}$, $N_{3}$, $N_{4}$), and the axes of rotation for the prisms ($R_{A}$, $R_{B}$) are expressed by:(3)$$L_{0}={\left[\begin{array}{ccc} 1 & 0 & 0 \end{array}\right]}^{T}$$
(4)$$N_{1}={\left[\begin{array}{ccc} 1 & 0 & 0 \end{array}\right]}^{T}$$
(5)$$N_{2}={\left[\begin{array}{ccc} \cos\left(\alpha \right) & -\sin\left(\Omega_{A}\right)\sin\left(\alpha \right) & \cos(\Omega_{A})\sin\left(\alpha \right) \end{array}\right]}^{T}$$
(6)$$N_{3}={\left[\begin{array}{ccc} \cos\left(\alpha \right) & \sin\left(\Omega_{B}\right)\sin\left(\alpha \right) & -\cos\left(\Omega_{B}\right)\sin\left(\alpha \right) \end{array}\right]}^{T}$$
(7)$$N_{4}={\left[\begin{array}{ccc} 1 & 0 & 0 \end{array}\right]}^{T}$$
(8)$$R_{A}={\left[\begin{array}{ccc} 1 & 0 & 0 \end{array}\right]}^{T}$$
(9)$$R_{B}={\left[\begin{array}{ccc} 1 & 0 & 0 \end{array}\right]}^{T}$$
where α is the wedge angle of the prisms.

The 3D vector form of Snell’s law, modified from [20], is expressed by:(10)$$L_{i+1}=\frac{n_{i}}{n_{i+1}}\left[L_{i}-\left(N_{i+1}\cdot L_{i}\right)N_{i+1}\right]+N_{i+1}\sqrt{1-{\left(\frac{n_{i}}{n_{i+1}}\right)}^{2}\left[1-{\left(N_{i+1}\cdot L_{i}\right)}^{2}\right]}$$
where $L_{i}$ and $L_{i+1}$ (for *i =* 0, 1, 2, 3) represent the unit vectors for the direction of the initial beam and the refracted beam, respectively. $N_{i+1}$ represents the unit vector for the direction of the planar normal on the refracting prism face, as expressed in Equations (4)–(7). $n_{i}$ and $n_{i+1}$ represent the refractive indices for the mediums (i.e., air or glass) on the initial and refracted sides of the prism face, respectively.

Once the unit vector for the direction of the emergent beam ($L_{4}$) has been determined, the equivalent azimuth and zenith angular observations—as defined with respect to the reference coordinate system of the Mid-40 (Figure 5)—are expressed by:(11)$$\phi =\arctan\left(\frac{L_{4y}}{L_{4x}}\right)$$
(12)$$\theta =\arccos\left(L_{4z}\right)$$
where ϕ and θ are the azimuth and zenith angular observations, respectively, and $L_{4x}$, $L_{4y}$, $L_{4z}$ are the components of the unit vector for the emergent beam.

### 2.2. Practical Angular Observation Model

Several studies [20,21,22] have been published that discuss the error sources and their effects on the angular accuracy of Risley prism-based observations. Li et al. [20] state that the systematic error sources of a Risley prism mechanism include component errors and assembly errors. The component errors include the errors in the wedge angle and refractive index for each prism. The assembly errors include the tilt errors, axis of rotation alignment errors, and rotational position (i.e., encoder) errors for each prism, and also the incident beam direction error.

In reality, some of the assumptions made within the ideal angular observation model do not hold true. Within the practical angular observation model it is still assumed that the prisms are geometrically identical and made of the same homogeneous medium. However, the assumption that the planar normals for the faces of each prism all lie in a common vertical plane is no longer valid. Rather, for each prism a unique plane exists that contains the prism’s two planar normals and defines the so-called *principal section* of the prism (Figure 6). The zero position for each prism (i.e., $\Omega_{A}\;\mathrm{and}\;\Omega_{B}=0{}^{\circ}$) is defined when its principal section is orientated vertically. Only by determining the principal section of each prism can the initial orientation of each prism be accurately calibrated and defined as the reference for calculating the prism rotation angle [20].

The reference coordinate system for the practical observation model is defined as a right-handed system with its origin at the intersection of the axis of rotation of prism B and perpendicular face of prism B. The axis of rotation of prism B ($R_{B}$) defines the X coordinate axis and the vertically oriented principal section of prism B contains the orthogonal Z coordinate axis (Figure 6). Equation (9) from the ideal observation model still correctly defines the unit vector for $R_{B}$. However, the axis of rotation for prism A ($R_{A}$) is no longer considered collinear with $R_{B}$, and is affected by the so-called *bearing tilt error*. As a result, the unit vector for $R_{A}$ is now expressed by:(13)$$R_{A}=\left[\begin{array}{c} \cos\left(\Delta \phi_{R_{A}}\right)\cos\left(\Delta \theta_{R_{A}}\right)\\ -\sin\left(\Delta \phi_{R_{A}}\right)\cos\left(\Delta \theta_{R_{A}}\right)\\ \sin\left(\Delta \theta_{R_{A}}\right) \end{array}\right]$$
where $\Delta \phi_{R_{A}}$ and $\Delta \theta_{R_{A}}$ represent the bearing tilt error for the axis of rotation of prism A within the azimuth (XY) plane and corresponding zenith (⊥ XY) plane, respectively.

The unit vector for the direction of the incident beam ($L_{0}$) is also no longer considered collinear with $R_{B}$, and is affected by the so-called *incident beam direction error*. As a result, the unit vector for $L_{0}$ is now expressed by:(14)$$L_{0}=\left[\begin{array}{c} \cos\left(\Delta \phi_{L_{0}}\right)\cos\left(\Delta \theta_{L_{0}}\right)\\ -\sin\left(\Delta \phi_{L_{0}}\right)\cos\left(\Delta \theta_{L_{0}}\right)\\ \sin\left(\Delta \theta_{L_{0}}\right) \end{array}\right]$$
where $\Delta \phi_{L_{0}}$ and $\Delta \theta_{L_{0}}$ represent the incident beam direction error within the azimuth (XY) plane and corresponding zenith (⊥ XY) plane, respectively.

The unit vectors for the direction of the planar normals for all four prism faces are affected by the so-called *prism tilt errors* for each respective prism. As a result, the unit vectors at the zero position of each prism ($N_{1}^{ο}$, $N_{2}^{ο}$, $N_{3}^{ο}$, $N_{4}^{ο}$) are expressed by:(15)$$N_{1}^{ο}=\left[\begin{array}{c} \cos\left(\Delta \phi_{R_{A}}+\Delta \phi_{A}\right)\cos\left(\Delta \theta_{R_{A}}+\Delta \theta_{A}\right)\\ -\sin\left(\Delta \phi_{R_{A}}+\Delta \phi_{A}\right)\cos\left(\Delta \theta_{R_{A}}+\Delta \theta_{A}\right)\\ \sin\left(\Delta \theta_{R_{A}}+\Delta \theta_{A}\right) \end{array}\right]$$
(16)$$N_{2}^{ο}=\left[\begin{array}{c} \cos\left(\Delta \phi_{R_{A}}+\Delta \phi_{A}\right)\cos\left(\Delta \theta_{R_{A}}+\Delta \theta_{A}+\alpha \right)\\ -\sin\left(\Delta \phi_{R_{A}}+\Delta \phi_{A}\right)\cos\left(\Delta \theta_{R_{A}}+\Delta \theta_{A}+\alpha \right)\\ \sin\left(\Delta \theta_{R_{A}}+\Delta \theta_{A}+\alpha \right) \end{array}\right]$$
(17)$$N_{3}^{ο}=\left[\begin{array}{c} \cos\left(\Delta \phi_{B}\right)\cos\left(\Delta \theta_{B}-\alpha \right)\\ -\sin\left(\Delta \phi_{B}\right)\cos\left(\Delta \theta_{B}-\alpha \right)\\ \sin\left(\Delta \theta_{B}-\alpha \right) \end{array}\right]$$
(18)$$N_{4}^{ο}=\left[\begin{array}{c} \cos\left(\Delta \phi_{B}\right)\cos\left(\Delta \theta_{B}\right)\\ -\sin\left(\Delta \phi_{B}\right)\cos\left(\Delta \theta_{B}\right)\\ \sin\left(\Delta \theta_{B}\right) \end{array}\right]$$
where $\Delta \phi_{A}$ and $\Delta \theta_{A}$ represent the tilt error within the azimuth (XY) plane and corresponding zenith (⊥ XY) plane for prism A, respectively, and $\Delta \phi_{B}$ and $\Delta \theta_{B}$ represent the tilt error within the azimuth (XY) plane and corresponding zenith (⊥ XY) plane for prism B, respectively.

The unit vectors for the direction of the planar normals for all four prism faces at the time of observation ($N_{1}$, $N_{2}$, $N_{3}$, $N_{4}$) can be determined by rotating the unit vectors at the zero position about the axis of rotation for each prism by the respective rotation angles $\Omega_{A}$ and $\Omega_{B}$. The rotation angles at the time of observation are defined by Equations (1) and (2). The unit vector for the axis of rotation for prism A ($R_{A}$) is non-trivial, as expressed in Equation (13), because $R_{A}$ suffers from bearing tilt error. As a result, a quaternion representation is used to perform the rotation of $N_{1}^{ο}$ and $N_{2}^{ο}$ about $R_{A}$ by the rotation angle $\Omega_{A}$. The $N_{1}$ and $N_{2}$ unit vectors are now expressed by Equations (23) and (24).
(19)$$q_{N_{1}^{ο}}=0+N_{1x}^{ο}\;i+N_{1y}^{ο}\;j+N_{1z}^{ο}\;k$$
(20)$$q_{N_{2}^{ο}}=0+N_{2x}^{ο}\;i+N_{2y}^{ο}\;j+N_{2z}^{ο}\;k$$
(21)$$q_{R_{A}}=\cos\left(\frac{\Omega_{A}}{2}\right)+\left(R_{Ax}\;i+R_{Ay}\;j+R_{Az}\;k\right)\cdot \sin\left(\frac{\Omega_{A}}{2}\right)$$
(22)$$q_{R_{A}}^{*}=\cos\left(\frac{\Omega_{A}}{2}\right)-\left(R_{Ax}\;i+R_{Ay}\;j+R_{Az}\;k\right)\cdot \sin\left(\frac{\Omega_{A}}{2}\right)$$
(23)$$N_{1}=q_{R_{A}}⊗q_{N_{1}^{ο}}⊗q_{R_{A}}^{*}$$
(24)$$N_{2}=q_{R_{A}}⊗q_{N_{2}^{ο}}⊗q_{R_{A}}^{*}$$
where q represents a unit quaternion and the terms **i**, **j**, **k** are the fundamental quaternion units, the terms $N_{1x}^{ο}$, $N_{1y}^{ο}$, $N_{1z}^{ο}$ are the components of the unit vector $N_{1}^{ο}$, the terms $N_{2x}^{ο}$, $N_{2y}^{ο}$, $N_{2z}^{ο}$ are the components of the unit vector $N_{2}^{ο}$, the terms $R_{Ax}$, $R_{Ay}$, $R_{Az}$ are the components of the unit vector $R_{A}$, and ⊗ represents the Hamilton product.

The axis of rotation for prism B ($R_{B}$) defines the X axis of the reference coordinate system, therefore the $N_{3}$ and $N_{4}$ unit vectors are simply expressed by:(25)$$N_{3}=\left[\begin{array}{ccc} 1 & 0 & 0\\ 0 & \cos\left(\Omega_{B}\right) & -\sin\left(\Omega_{B}\right)\\ 0 & \sin\left(\Omega_{B}\right) & \cos\left(\Omega_{B}\right) \end{array}\right]\cdot N_{3}^{ο}$$
(26)$$N_{4}=\left[\begin{array}{ccc} 1 & 0 & 0\\ 0 & \cos\left(\Omega_{B}\right) & -\sin\left(\Omega_{B}\right)\\ 0 & \sin\left(\Omega_{B}\right) & \cos\left(\Omega_{B}\right) \end{array}\right]\cdot N_{4}^{ο}$$

Once the unit vectors for the directions of the planar normals at the time of observation have been defined, the unit vector for the direction of the emergent beam ($L_{4}$) can be determined. Equations (14) and (23) are substituted into Equation (10) which determines the unit vector for the direction of the first refracted beam ($L_{1}$). Equation (10) is applied three more times by substituting in Equations (24)–(26) to determine the unit vectors $L_{2}$, $L_{3}$, and $L_{4}$, respectively. Lastly, Equations (11) and (12) are used to convert the unit vector $L_{4}$ into the equivalent azimuth and zenith angular observations.

Within the practical angular observation model a total of 13 model parameters are needed to determine the azimuth and zenith observations for the emergent beam at the time of observation. A time of observation equal to zero corresponds to the prisms being in their zero positions, as previously discussed. The model parameters are summarized in Table 2.

### 2.3. Observability of the Model Parameters

Liu et al. [16] state that the Mid-40 is composed of two identical prisms with a refractive index of 1.51, wedge angles of 18°, and angular velocities of −4664 revolutions per minute (RPM) and 7294 RPM for prisms A and B, respectively, following a right-hand rule for positive rotation around the X coordinate axis. However, the refractive index of air, the refractive index of the prisms, and the wedge angle of the prisms model parameters are perfectly correlated due to the use of Snell’s law within the observation model. As a result, the refractive index of air is set to a constant of 1.00 and the wedge angle of the prisms is set to a constant of 18° within the observation model. The *refractive index of the prisms* model parameter is not fixed to a constant value to allow it to be optimized based on the sensor’s observations.

The azimuth and zenith angular observations—which are reported to a precision of 0.01°—are the raw outputs available from the Mid-40 sensor. It is important to analyze the sensitivity of the output observations with respect to changes in the model parameters, as well as the correlations between the model parameters, in order to understand the observability of the model. Using an extended Kalman filter—as described in the next section—analyses were performed to study the nature of these sensitivities and correlations. It was discovered that changes in all of the model parameters, except for the *horizontal tilt error for prism A* ($\Delta \phi_{A}$), were observable with respect to changes in the azimuth and zenith observations. The *horizontal tilt error for prism A* demonstrated to be relatively insensitive to changes in the azimuth and zenith observations and indicated a strong negative correlation (*p* = −0.99) with the computed estimates for the *rotation angle for prism A* ($\Omega_{A}$) and the *vertical tilt error for prism A* model parameter ($\Delta \theta_{A}$). As a result, the *horizontal tilt error for prism A* model parameter was assumed to be zero, and therefore had no effect on the practical observation model and was removed.

### 2.4. Estimation of the Model Parameters and Prism Rotation Angles Using a Kalman Filter

The practical angular observation model is utilized within an EKF to estimate the model parameters and the rotation angles of each prism at the observation times. The EKF is processed in both the forward and backward time directions, and the respective results are combined to produce an optimal smoothed estimate [23]. The EKF dynamics model for the system is expressed by:(27)$$\left[\begin{array}{c} \dot{n_{prism}}\\ \dot{\omega_{A}}\\ \dot{\omega_{B}}\\ \dot{\Delta \phi_{L_{0}}}\\ \dot{\Delta \theta_{L_{0}}}\\ \dot{\Delta \phi_{R_{A}}}\\ \dot{\Delta \theta_{R_{A}}}\\ \dot{\Delta \theta_{A}}\\ \dot{\Delta \phi_{B}}\\ \dot{\Delta \theta_{B}}\\ \dot{\Omega_{A}}\\ \dot{\Omega_{B}} \end{array}\right]=\left[\begin{array}{c} 0\\ 0\\ 0\\ 0\\ 0\\ 0\\ 0\\ 0\\ 0\\ 0\\ \omega_{A}\\ \omega_{B} \end{array}\right]+\left[\begin{array}{c} w_{1}\\ w_{2}\\ w_{3}\\ w_{4}\\ w_{5}\\ w_{6}\\ w_{7}\\ w_{8}\\ w_{9}\\ w_{10}\\ 0\\ 0 \end{array}\right]$$
where $w_{i}=𝓝\left(0,\sigma_{w_{i}}^{2}\right)$ and represents the state transition noise.

The EKF measurement model for the system is highly non-linear and requires the use of Equations (13)–(26) and four successive applications of Equation (10) before the following measurement equations can be utilized:(28)$$\phi =\arctan\left(\frac{L_{4y}}{L_{4x}}\right)+v_{1}$$
(29)$$\theta =\arccos\left(L_{4z}\right)+v_{2}$$
where $v_{i}=𝓝\left(0,\sigma_{v_{i}}^{2}\right)$ and represents the measurement noise.

The EKF is initialized by examining the output azimuth and zenith observations and finding a time of observation that indicates the prisms are approximately in their zero positions. This occurs when the azimuth observation is approximately 0° and the zenith observation is approximately at its maximum value (109.2°). This time of observation is set equal to zero and all observations preceding this instance are disregarded and all proceeding observation times are adjusted accordingly. The EKF state variables are initialized as:(30)$$\left[\begin{array}{c} n_{prism}\\ \omega_{A}\\ \omega_{B}\\ \Delta \phi_{L_{0}}\\ \Delta \theta_{L_{0}}\\ \Delta \phi_{R_{A}}\\ \Delta \theta_{R_{A}}\\ \Delta \theta_{A}\\ \Delta \phi_{B}\\ \Delta \theta_{B}\\ \Omega_{A}\\ \Omega_{B} \end{array}\right]=\left[\begin{array}{c} 1.51\\ -27,984\;{}^{\circ}/s\\ 43,764\;{}^{\circ}/s\\ 0{}^{\circ}\\ 0{}^{\circ}\\ 0{}^{\circ}\\ 0{}^{\circ}\\ 0{}^{\circ}\\ 0{}^{\circ}\\ 0{}^{\circ}\\ 0{}^{\circ}\\ 0{}^{\circ} \end{array}\right]$$

It was discovered that a Mid-40 sensor—installed with v03.08.0000 firmware—operates using two different combinations of angular velocities for the prisms. The first combination uses nominal values of −27,984 °/s (—4664 RPM) and 43,764 °/s (7294 RPM) for $\omega_{A}$ and $\omega_{B}$, respectively, while the second combination uses nominal values of −43,764 °/s and 27,984 °/s for $\omega_{A}$ and $\omega_{B}$, respectively. The combinations alternate in occurrence when the sensor is power cycled, and it is speculated that this increases the lifespan of the brushless direct current (BLDC) motors that rotate the prisms. Therefore, the initial values for the 2nd and 3rd vector elements within Equation (30) must change when the second combination of angular velocities is experienced during data collection.

### 2.5. Estimation of the Model Parameters Using a Planar Constrained Least-Squares Adjustment

The practical angular observation model is also utilized within a non-linear least-squares adjustment to estimate the model parameters—excluding the angular velocities and the *refractive index of the prisms*—based on constraining the resulting Cartesian coordinates of the observed points to best-fit a planar surface. The point coordinates are calculated within the reference coordinate system of the Mid-40 using Equation (31):(31)$$\left[\begin{array}{c} x_{p}\\ y_{p}\\ z_{p} \end{array}\right]=\left[\begin{array}{c} r_{p}\cdot \sin\left(\theta_{p}\right)\cdot \cos\left(\phi_{p}\right)\\ r_{p}\cdot \sin\left(\theta_{p}\right)\cdot \sin\left(\phi_{p}\right)\\ r_{p}\cdot \cos\left(\theta_{p}\right) \end{array}\right]$$
where $x_{p},y_{p},z_{p}$ represent the Cartesian coordinates of point p, and $r_{p},\phi_{p},\theta_{p}$ represent the range, azimuth, and zenith observations from a Mid-40 sensor to point p, respectively.

Before the adjustment begins, the input azimuth and zenith observations are processed using the EKF—as previously discussed—to estimate the current model parameters and the instantaneous rotation angles of the prisms that correspond to the input observations. Each iteration of the least-squares adjustment begins with computing new estimates for the azimuth and zenith angular observations for all points, using the practical angular observation model and the current estimates for the model parameters and the prism rotation angles. The point coordinates are then calculated and used within a singular value decomposition (SVD) problem to estimate the current best-fitting plane based on all the point coordinates. The perpendicular distances between the points and the best-fit plane are then calculated and used to define the minimization criteria for the least-squares adjustment. Equation (32) illustrates the unweighted least-squares solution:(32)$$\Delta ={\left(A^{T}A\right)}^{-1}A^{T}\varepsilon$$
where Δ represents the corrections to the current estimates for the model parameters, A represents the Jacobian matrix containing the partial derivatives of the point-to-plane distances with respect to the model parameters, and ε represents the vector containing the negated values of the point-to-plane distances. The solution is repeated until convergence of the standard deviation of unit weight is achieved.

## 3. Results

### 3.1. Analysis of Real Mid-40 Angular Observations

Angular observations were collected from a stationary Mid-40 sensor and used within the EKF to estimate the observation model parameters, the prism rotation angles at the observation times, and the azimuth and zenith observation residuals. The estimated observation model parameters are considered to be an accurate representation of the unique calibration parameters of the sensor, as determined by the manufacturer and stored within the sensor’s non-volatile memory. The sensor was connected to a pulse-per-second (PPS) timing signal from a GNSS receiver to ensure accurate time observations were collected. Due to the fast 100 kHz observation rate of the Mid-40, only a small dataset containing 30 s of observations was used. To minimize the impact of any autocorrelations within the observations only one pair of azimuth and zenith observations for every one hundred collected pairs were used within the EKF (i.e., a 1 kHz observation rate was used). Figure 7 and Figure 8 illustrate the estimated mean values and sample distributions for the observation model parameters based on ~30,000 optimal smoothed estimates from the EKF (i.e., the output of each predict-measure-update iteration of the filter).

For a stationary sensor it was observed that the angular velocities of the prisms oscillate over time with approximately sinusoidal responses. The oscillations are speculated to be a result of the type, quality, and logic of the speed controllers used within the Mid-40’s electronics to control the BLDC motors that rotate each of the prisms [24]. Figure 9 illustrates the EKF smoothed estimates for the angular velocities of prisms A and B during the data collection period. The EKF also estimated the residuals for the azimuth and zenith observations and the respective distributions were approximately normal with means of 0° and standard deviations of 0.004° (Figure 10). The skewness and kurtosis values for the azimuth and zenith residual distributions were 0.00 and 0.64, and 0.01 and 0.59, respectively.

### 3.2. Analysis of Simulated Mid-40 Angular Observations

Using the practical angular observation model—with the same set of model parameters as estimated for the real Mid-40 sensor in the previous section—azimuth and zenith observations were computed using 30 s of 1 kHz prism rotation angles. Randomly sampled errors from a normal distribution with a mean of 0° and a standard deviation of ±0.01° were added to the azimuth and zenith observations to produce a simulated dataset for a Mid-40 sensor. The simulated dataset was processed using the EKF to analyze if the model parameters, prism rotation angles, and observation errors could be correctly estimated. The results of the analysis demonstrated that all of the terms of interest could be accurately and precisely estimated. Table 3 lists the known terms used within the simulation and their corresponding EKF estimates. Figure 11 illustrates the sample distributions for the estimated rotation angle residuals for prism A and prism B based on the simulated observations.

### 3.3. Analysis of Simulated Mid-40 Observations from an Incorrectly Calibrated Sensor

A final analysis was performed that simulated the scenario where the unchangeable manufacturer’s calibration values (i.e., model parameters) for a Mid-40 sensor were no longer correct. This may be the result of the sensor experiencing significant vibrations which cause physical changes to the alignment of the internal components within the sensor. In this case, any future angular observations made with the sensor would be inaccurate, but the range observations would be practically unaffected. The goal of the analysis was to investigate if range observations made to a planar surface could be utilized to estimate the correct calibration values for a sensor and therefore improve the accuracy of the angular observations.

Using the practical angular observation model along with 10 s of 1 kHz prism rotation angles, azimuth and zenith observations were computed using two different sets of model parameters. The first set of model parameters were the same as those estimated for the real Mid-40 sensor in Section 3.1 and defined the correct calibration. The second set of model parameters had all the model’s error parameters equal to 0° and represented the manufacturer’s unchangeable calibration values (i.e., the incorrect calibration). The correctly calibrated azimuth and zenith observations were used to compute simulated range observations from a Mid-40 sensor to a known planar surface. The planar surface was simulated to be 30 m away from the sensor with its planar normal being misaligned by 10° horizontally and 10° vertically with respect to the sensor’s optical scan axis. It was discovered that the observed planar surface needed to be non-perpendicular to the optical scan axis in order to break the rotational symmetry of the circular FoV. The simulated dataset consisted of the incorrectly computed azimuth and zenith observations along with the correctly computed range observations.

Before running the analysis, randomly sampled errors from normal distributions with means of 0° and 0 mm and standard deviations of ±0.01° and ±20 mm were added to the simulated azimuth and zenith observations, and the range observations, respectively. The root mean square error (RMSE) measures for the simulated azimuth and zenith observations were ±0.077° and ±0.396°, respectively. The simulated dataset was processed using the planar constrained least-squares adjustment and the results demonstrated that the model parameters—and therefore the azimuth and zenith observations—could be estimated with improved accuracy and precision. The adjusted azimuth and zenith observations reported improved RMSE measures of ±0.066° and ±0.022°, respectively. However, biases of −0.065° and −0.021° were still present within the angular observation residuals and are speculated to be a result of the range observation uncertainty limiting the achievable accuracy of the adjustment. Therefore, the 20 mm range uncertainty of the Mid-40 sensor defines the limit to the improvement of the angular observations using this method. Figure 12 illustrates the sample distributions for the original and adjusted angular observation residuals.

## 4. Discussion and Conclusions

Within the analyses of this study, real and simulated angular observations for a Mid-40 sensor were used to demonstrate that the practical angular observation model is accurate and precise, and enables rigorous uncertainty estimation of the angular observations from a Risley prism beam steering mechanism to be performed. The residuals between the model’s angular observations and those from an actual Mid-40 sensor were shown to be normally distributed with a mean of 0° and a standard deviation of ±0.004°. This suggests that the angular observations for the Mid-40 sensor are more accurate than the manufacturer’s specification of <0.1°. However, this finding does not take into consideration the angular uncertainty associated with the direction of the emergent beam due to beam divergence. The previous studies [18,19] provided inferred and qualitative evidence that the angular accuracy of the Mid-40 sensor was in agreement with the manufacturer’s specification, but also stated that the lack of information regarding the internal hardware configuration and operating principles of the sensor restricted further analysis. This study was performed to overcome these restrictions, by introducing an angular observation model and providing a rigorous accuracy assessment of the Mid-40’s angular observations.

Using the angular observation model within an extended Kalman filter allows for the azimuth and zenith observations to be used to estimate the model parameters as well as a set of lower-level observations, namely the rotation angles for each prism. The filter can accurately estimate the model parameters and the rotation angles for the prisms even when the prisms’ angular velocities are not constant. The inherent benefit of the prism rotation angles is that they are not affected by the systematic component errors and assembly errors of the sensor. By using the prism rotation angles and the corresponding range observations, an independent estimation of the sensor’s calibration values can be performed, and provides a method for improving the angular observations of an incorrectly calibrated Mid-40 sensor. However, the uncertainty in the range observations does set a limit to the achievable accuracy of the independent estimation.

In addition to the systematic errors that affected the angular observations, earlier research conducted as part of this study included analyzing errors within the range observation and the Cartesian coordinate representation of the azimuth, zenith, and range observations. No obvious systematic errors within the range observation could be identified; if any errors remain, they are likely non-geometric in nature and small in magnitude. Within the Cartesian coordinate representation of the observations a systematic error was discovered, and was a result of the emergent beam not coinciding with the origin of the sensor’s reference coordinate system. The error caused the Y and Z coordinates to be incorrect by upwards of 4.5 mm, see the Appendix A for details. Future work will include (1) conducting experiments using real Mid-40 angular and range observations to planar surfaces and estimating an improved set of calibration values for a sensor and quantifying that improved angular observations can be realized; (2) analyzing the angular observations from a triple Risley prism lidar sensor (e.g., Livox Horizon and Livox); and (3) studying the spatial distribution properties of a point cloud generated from a UAS lidar mapping system that utilizes a Mid-40 sensor.

## Figures and Tables

**Figure 1 sensors-21-04722-f001:**
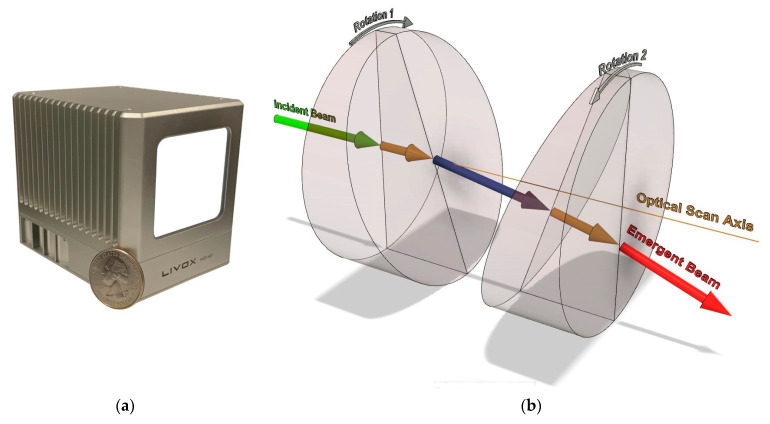
(**a**) Livox Mid-40 lidar sensor with a U.S. quarter dollar for scale; (**b**) Risley prism optical steering mechanism showing two wedge-shaped glass prisms, the incident beam, three intermediate refracted beams, the emergent beam, and the optical scan axis.

**Figure 2 sensors-21-04722-f002:**
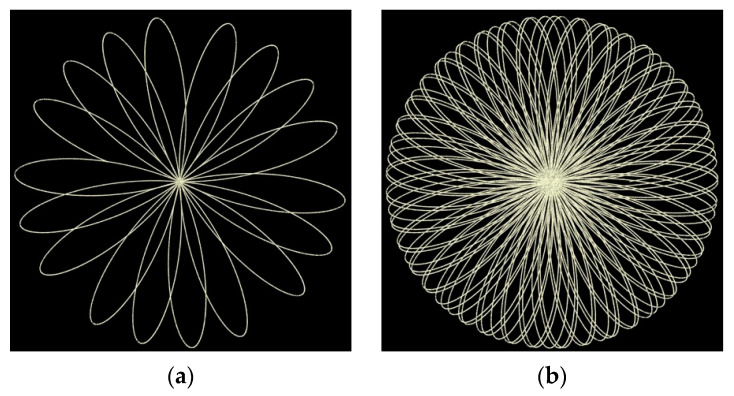
(**a**) Mid-40 rosette scan pattern after 0.09 s; (**b**) Observation density within the circular field of view of the Mid-40 after 0.5 s, notice the increase in density near the center.

**Figure 3 sensors-21-04722-f003:**
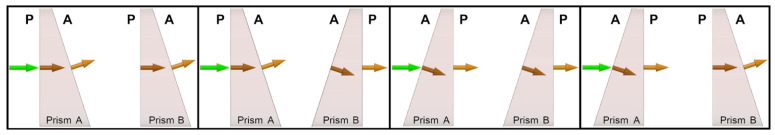
Four configurations for a Risley prism steering mechanism showing the two wedge-shaped glass prisms, the incident beam, and the planar normal for each prism face.

**Figure 4 sensors-21-04722-f004:**
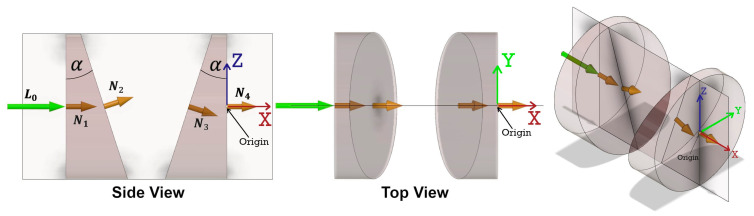
An ideal Risley prism PA-AP configuration in the zero position of the prisms at time (*t*) = 0. The incident beam and the planar normals for the prism faces all lie in a common vertical plane. The right-handed X, Y, Z (red, green, blue) Cartesian reference coordinate system is shown.

**Figure 5 sensors-21-04722-f005:**
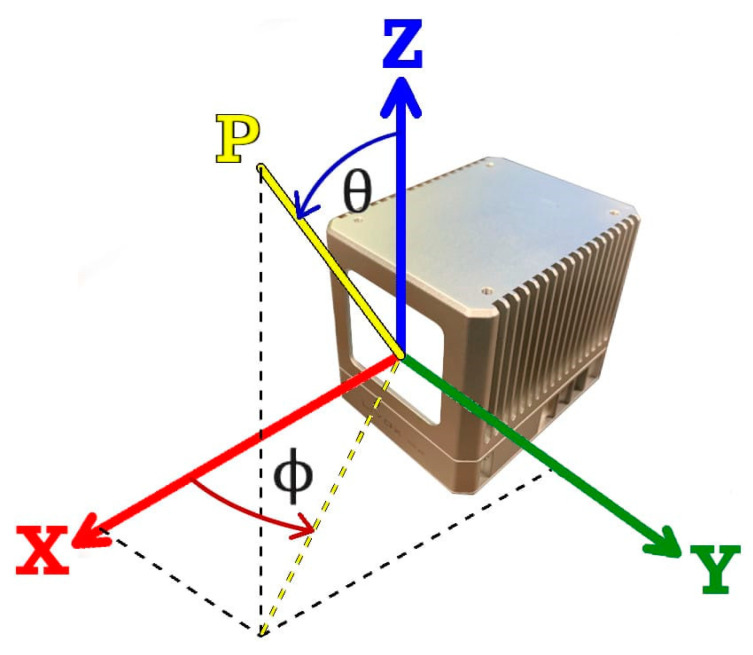
Definition of the azimuth (ϕ) and zenith (θ) angular observations to point P for the Livox Mid-40 lidar sensor with respect to its Cartesian reference coordinate system.

**Figure 6 sensors-21-04722-f006:**
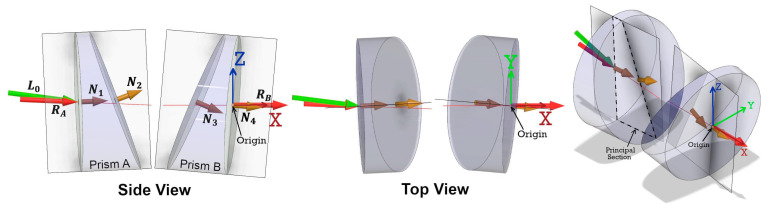
A practical Risley prism PA-AP configuration in the zero position of the prisms. The two planar normals for each prism lie a unique plane—containing the principal section—and when orientated vertically define the zero position for each prism. Each prism is affected by tilt errors and has a unique axis of rotation ($R_{A}$, $R_{B}$ ). The incident beam is no longer collinear with the optical scan axis. The right-handed X, Y, Z (red, green, blue) Cartesian reference coordinate system is shown.

**Figure 7 sensors-21-04722-f007:**
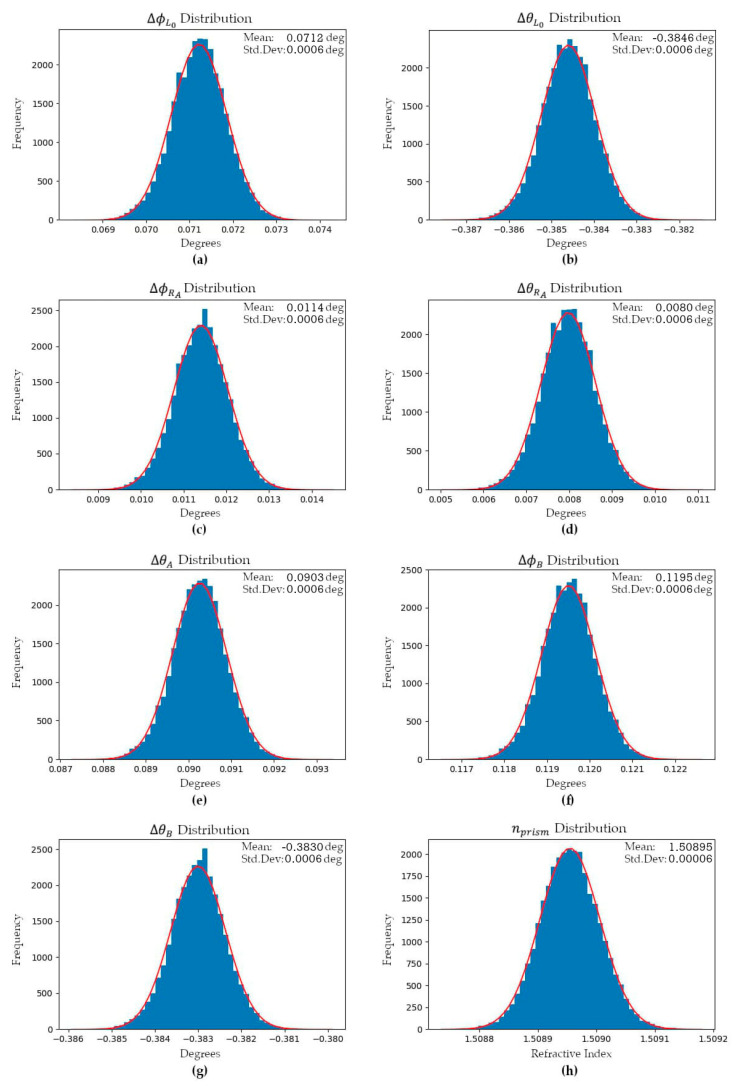
Sample distributions for the model parameters: (**a**) horizontal incident beam direction error; (**b**) vertical incident beam direction error; (**c**) horizontal bearing tilt error for prism A; (**d**) vertical bearing tilt error for prism A; (**e**) vertical tilt error for prism A; (**f**) horizontal tilt error for prism B; (**g**) vertical tilt error for prism B; (**h**) refractive index of prisms.

**Figure 8 sensors-21-04722-f008:**
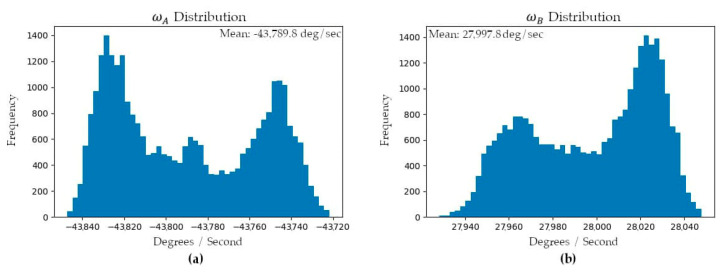
Sample distributions for the model parameters: (**a**) angular velocity of prism A; (**b**) angular velocity of prism B.

**Figure 9 sensors-21-04722-f009:**
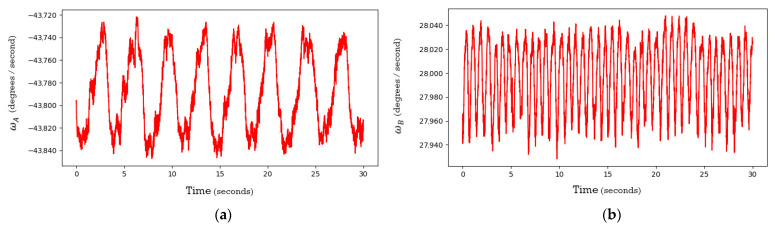
Time plot of the angular velocity of: (**a**) prism A; (**b**) prism B.

**Figure 10 sensors-21-04722-f010:**
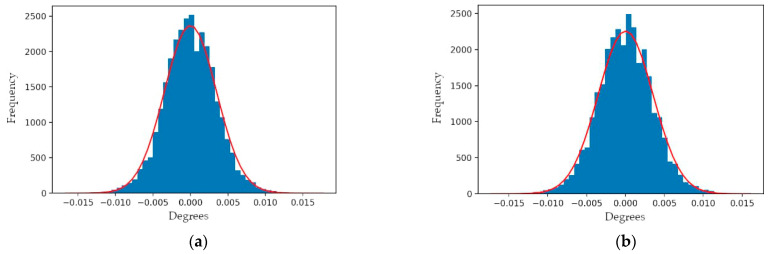
(**a**) Sample distribution for residuals of: (**a**) real azimuth observations; (**b**) real zenith observations.

**Figure 11 sensors-21-04722-f011:**
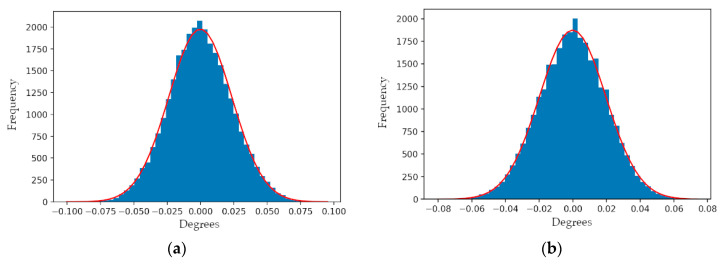
Sample distribution for the simulated rotation angle residuals of: (**a**) prism A; (**b**) prism B.

**Figure 12 sensors-21-04722-f012:**
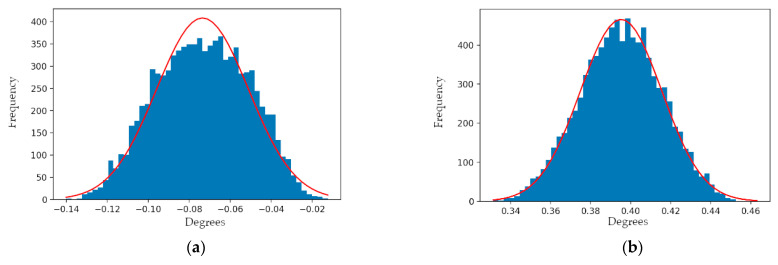
Sample distribution for the: (**a**) pre-adjustment azimuth observation residuals; (**b**) pre-adjustment zenith observation residuals; (**c**) post-adjustment azimuth observation residuals; (**d**) post-adjustment zenith observation residuals, for the simulated dataset containing noise. Note the change in abscissa scales between pre- and post-adjustment plots.

**Table 1 sensors-21-04722-t001:** Manufacturer’s Specifications for the Livox Mid-40 Lidar Sensor (Source: [17]).

Specification	Value
Laser Wavelength	905 nm
Laser Safety	Class 1 (IEC 60825-1:2014) eye safe
Detection Range	90 m @ 10% reflectivity 130 m @ 20% reflectivity 260 m @ 80% reflectivity
Field of View	38.4° (Circular)
Range Precision	20 mm (1σ @ 20 m)
Angular Accuracy	<0.1°
Beam Divergence	0.28° (Vertical) × 0.03° (Horizontal)
Point Rate	100,000 points/s
False Alarm Rate	<0.01%
Weight (with cable)	760 g
Dimensions	88 mm × 76 mm × 69 mm
Operating Temperatures	−20 °C to 65 °C
Water and Dust Rating	IP67
Price	$599 USD

**Table 2 sensors-21-04722-t002:** Parameters for the Practical Angular Observation Model.

Parameter Name	Symbol
Refractive Index of Air	$n_{air}$
Wedge Angle of Prisms	α
Refractive Index of Prisms	$n_{prism}$
Angular Velocity of Prism A	$\omega_{A}$
Angular Velocity of Prism B	$\omega_{B}$
Horizontal Incident Beam Direction Error	$\Delta \phi_{L_{0}}$
Vertical Incident Beam Direction Error	$\Delta \theta_{L_{0}}$
Horizontal Bearing Tilt Error for Prism A	$\Delta \phi_{R_{A}}$
Vertical Bearing Tilt Error for Prism A	$\Delta \theta_{R_{A}}$
Horizontal Tilt Error for Prism A	$\Delta \phi_{A}$
Vertical Tilt Error for Prism A	$\Delta \theta_{A}$
Horizontal Tilt Error for Prism B	$\Delta \phi_{B}$
Vertical Tilt Error for Prism B	$\Delta \theta_{B}$

**Table 3 sensors-21-04722-t003:** Comparison of simulated known and estimated terms.

Term of Interest	Simulated Known Value	EKF Estimated Value
Refractive Index of Prisms	1.5090	1.5090±0.0001
Angular Velocity of Prism A	−43,789.8 °/s	−43,789.8 °/s±2.2 °/s
Angular Velocity of Prism B	27,997.8 °/s	27,997.8 °/s±2.2 °/s
Horizontal Incident Beam Direction Error	0.071°	0.071°±0.002°
Vertical Incident Beam Direction Error	−0.385°	−0.385°±0.002°
Horizontal Bearing Tilt Error for Prism A	0.011°	0.013°±0.002°
Vertical Bearing Tilt Error for Prism A	0.008°	0.010°±0.002°
Vertical Tilt Error for Prism A	0.090°	0.089°±0.002°
Horizontal Tilt Error for Prism B	0.120°	0.117°±0.002°
Vertical Tilt Error for Prism B	−0.383°	−0.384°±0.002°
Azimuth Observation Errors	0°±0.01°	0°±0.008°
Zenith Observation Errors	0°±0.01°	0°±0.008°
Rotation Angle Errors for Prism A	0°	0°±0.024°
Rotation Angle Errors for Prism B	0°	0°±0.020°

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
