# Peer review of "A Rigorous Observation Model for the Risley Prism-Based Livox Mid-40 Lidar Sensor"

_sensors, 2021, doi:10.3390/s21144722_

Round 1

Reviewer 1 Report

The article: “A Rigorous Observation Model for the Risley Prism-based Livox Mid-40 Lidar Sensor« deals with geometrical calibration of low cost lidar sensor. This type of sensors is gaining importance for use in autonomous vehicles, therefore the topic is highly relevant.

Two articles about Livox sensor calibration have already been published, both focusing mainly on distance measurements and reporting difficulties in angular component of measurements. The present article bridges the gap.

The physical model of operating sensor is thoroughly presented. First for theoretically perfect situation and then all possible structure imperfections influencing the output are presented mathematically. The mathematical model of the whole system is used in the extended Kalman filter allowing authors to perform calibration of actual sensor as well as simulate erroneous measurement with known error sources. The latter allows for the assessment of measurement quality with unknown or even wrong calibration parameters.

The article is well structured and written concisely. Abstract summarizes key information about the workflow and results. Adequate and relevant literature is cited throughout the article.

Well written article that requires no further improvements before publishing.

Particular comment:

In Figure 7: Mean and Std.Dev are given in each image with 3 decimal places. Even if the fourth may not be important, from the values on the x axis we can see that the peak is not at the 0.071 but somewhere between 0.071 and 0.072 (speaking for image (a) for example).

Author Response

Figure 7 has been updated to show an additional decimal place within the mean and std.dev. estimates.

Reviewer 2 Report

The paper is well written and the experiments to measure the angular error properties are presented clearly.

One minor suggestion is to move the findings presented in the appendix as a section in the main article - the Cartesian offset of the point cloud would seem to be rather important to potential users of the sensor. However, I will defer to the preference of the authors for the placement of this section.

On line 100, "artefact" should be changed to "artifact".

Author Response

The authors prefer to leave the Cartesian offset information within the appendix.

The spelling of artefact has been changed to artifact on lines 87, 88, and 100.

Reviewer 3 Report

This article refers about observation model for the Risley prism.

row 30: unoccupied aerial systems (UAS) [2] ...  it should be unmanned aerial system, but better is RPAS (remotely piloted aircraft system - it means, based on legislative, that there is a persaon, which is responsible for the drone)

The Material and Methods paragraph is too long; it should be divided on real material description and methods - data and mesurement processing; a flowchart will be nice.

row 112  ... write some words, why do you use just Kalman filter

equation (27) here I also recommend an abbreviated notation (matrix)

equation (32) ...may be, better (AT P A)   ...P is weighting factor (matrix)

References - add some references in introduction

autonomous vehicle navigation + UAS (drones)  ...only 2 references is too low; you can find thousands articles...

https://ieeexplore.ieee.org/abstract/document/5174729

https://www.tandfonline.com/doi/full/10.1080/22797254.2018.1527661

https://www.tandfonline.com/doi/full/10.1080/22797254.2019.1683471

references to TLS are missing (mainly on mobile or personal mobile laser scanners), they use a rotating systems (Zeb-Revo, Leica backpack, Greenvalley etc.  Find articles ,for example:

https://www.researchgate.net/publication/259770253_Possibilities_of_a_Personal_Laser_Scanning_System_for_Forest_Mapping_and_Ecosystem_Services

https://www.mdpi.com/2076-3417/11/4/1712

reference 3 is from apple page, ok, but you can find better references focused on reserch

https://www.researchgate.net/publication/327669935_Best_Practices_in_the_Use_of_Augmented_and_Virtual_Reality_Technologies_for_SLA_Design_Implementation_and_Feedback

In conclusion - I recomend add a clear benefit of your reserch - for example using on real technology, measurement refinement, etc.

Author Response

The University of Florida has an initiative to use gender-neutral terms whenever possible, therefore the term unmanned aerial systems is considered to be inappropriate. However, the term UAS is still widely used within existing literature and therefore the authors believe the term unoccupied aerial systems is appropriate. However, the term remotely piloted aircraft systems (RPAS) has been added to the manuscript in conjunction with the definition of UAS.

It is assumed that the referred to 'Materials and Methods paragraph' is referring to the entire 'Materials and Methods section' of the manuscript. The authors believe the methods employed within the research are described in minimally sufficient detail within the section and therefore the section can not be further reduced without sacrificing necessary details.

The authors have included additional reasoning for the use of the Kalman filter within the manuscript near row 112.

The authors believe the use of the differential equation in Equation 27 best illustrates to the reader the individual parameters within the state vector and their dynamic relationships. If an abbreviated equation was to be used additional explanation would be required or the reader would need to infer the dynamic relationships between the parameters.

The authors have clarified that an unweighted least-squares adjustment is being used and therefore the inclusion of the weight matrix (P) is not required within Equation 32.

Additional references for the use of lidar technology within autonomous vehicles, UAS mapping, ground-based personal mapping systems, and for generating augmented reality (AR) data and overlays have been included within the manuscript. In addition, more appropriate AR references are now cited. The authors agree with and appreciate the reviewer's recommendation.

A clear research benefit statement has been added to the conclusions section of the manuscript.